# A Survey of Vaping Use, Perceptions, and Access in Adolescents from South-Central Texas Schools

**DOI:** 10.3390/ijerph20186766

**Published:** 2023-09-15

**Authors:** Bretton A. Gilmore, Corbyn M. Gilmore, Kelly R. Reveles, Jim M. Koeller, Jodi H. Spoor, Bertha E. Flores, Christopher R. Frei

**Affiliations:** 1College of Pharmacy, The University of Texas at Austin, San Antonio, TX 78229, USA; gilmorec1@uthscsa.edu (C.M.G.); revelesk@uthscsa.edu (K.R.R.); koeller@uthscsa.edu (J.M.K.); 2Joe R. and Teresa Lozano Long School of Medicine, University of Texas Health San Antonio, San Antonio, TX 78229, USA; floresb2@uthscsa.edu; 3Graduate School of Biomedical Sciences, University of Texas Health San Antonio, San Antonio, TX 78229, USA; 4Audie L. Murphy Veterans Hospital, South Texas Veterans Health Care System, San Antonio, TX 78229, USA; 5University Hospital, San Antonio, TX 78229, USA; 6Southside Independent School District, San Antonio, TX 78221, USA; jodi.spoor@southsideisd.org; 7School of Public Health, University of Texas Health Houston, San Antonio, TX 78229, USA

**Keywords:** electronic cigarettes, vaping, adolescents, Texas, United States

## Abstract

Despite efforts to dissuade major manufacturers and retailers from marketing and selling vape products to adolescents, the practice of vaping continues to increase in this population. Few studies have assessed adolescent perceptions of vaping, access to vaping, and use of vaping, and most rely, at least in part, on inferential conclusions drawn from data on smoking traditional combustible cigarettes. A novel electronic survey was created to assess the use of vapes, perceptions of vaping, and access to vaping among a convenience sample of adolescents (ages 12–20 years) in eleven schools in South-Central Texas from May to August 2021. The students’ perceived threat of negative health outcomes due to vaping was calculated based on questions soliciting perceptions of severity (perceived danger) and susceptibility (perceived likelihood of illness). Trends were identified using descriptive and bivariate statistical tests. A total of 267 respondents were included; 26% had tried vaping. A majority (63%) did not believe vaping and smoking were synonymous. Most (70%) thought it was easy to obtain supplies and (76%) vape before and after (88%) or even during (64%) school. Respondents who vaped had a 34% lower perceived threat when compared to respondents who did not vape. In this sample of adolescents from South-Central Texas, one in four reported that they had tried vaping. Easy access to vapes and misperceptions regarding the safety of vaping might create a false sense of security with respect to vaping as an alternative to smoking, particularly among those who reported vaping, and is likely contributing to the increased use of vapes.

## 1. Introduction

Among adolescents, the use of e-cigarettes, commonly referred to as vaping, poses significant health risks [1]. Beginning in 2019, federal law prohibited the sale of tobacco products (including cigarettes and vapes) to persons under the age of 21 years. In Texas, selling tobacco products to underage adolescents is a Class C Misdemeanor. Despite federal and state age limit laws, it is estimated that five million youth vape [2,3]. One prior study reported that approximately one-quarter of high school students reported vaping within the past 30 days [4]. A review conducted by the 2018 National Academies of Sciences, Engineering, and Medicine concluded that adolescent vaping is strongly associated with the future use of traditional cigarettes [5]. Furthermore, vaping is an addictive activity that can lead to a dependence on nicotine that can often progresses to tetrahydrocannabinol (THC) abuse [6,7,8]. While most of the research remains focused on high school students, vaping is becoming increasingly popular in middle school, and adolescents are experimenting at earlier ages [9,10].

Numerous studies have described systemic adverse health effects due to vaping. A 2020 systematic review by Tzorti et al. [11] described reports of respiratory, allergic, gastrointestinal, cardiovascular, and hematologic effects, poisoning, and traumatic adverse effects associated with vaping. Importantly, the median age of these cases was 23 years, with underrepresentation of adolescents. Additionally, few studies have evaluated adolescent vaping knowledge, attitudes, and beliefs and most assumptions about adolescent perceptions and expectations have been predicated on data obtained from modified assessments designed to evaluate the use of combustible cigarettes [12]. Perceptions of health, or conversely, perceptions of threats to health, can be key indicators of risk tolerance or risk aversion in adolescents [13,14]. Adolescents primarily express that the reasons they begin vaping are due to curiosity, pleasure, and taste [15]. In the early years of vaping, many adolescents and young adults were using e-cigarettes to experiment with a multitude of flavors [16]. Moreover, adolescents’ perceptions of harm were shown to be directly tied to the flavor; fruity or candied flavors were seen as less harmful than tobacco flavors [17]. Adolescents that perceive the threat of relative harm to be low are more vulnerable to future use than abstainers with higher degrees of awareness regarding prospective dangers [18]. Misperceptions regarding the safety of vaping might create a false sense of security regarding vaping as an alternative to smoking. A lack of knowledge and/or attitudes and beliefs regarding the risks or potential health outcomes associated with vaping likely lead to uninformed decision making and risky habitual behaviors.

While policies and laws may deter adolescent vaping [19], further work is needed to identify use patterns and behaviors that may inform future public health interventions. Thus, the objective of this study was to evaluate the use of vapes, perceptions of vaping, and access to vapes in a sample of adolescents in South-Central Texas schools. This study hypothesized that adolescents might perceive vaping differently than traditional tobacco use or smoking combustible cigarettes.

## 2. Materials and Methods

### 2.1. Study Design

This was a study designed to collect information about the use of vaping, perceptions of vaping, and access to vaping among adolescents, using a novel survey. The study used a cross-sectional convenience sample design of adolescents in the greater San Antonio, Texas, area from 5 May 2021, to 3 August 2021. The study reporting is consistent with the Strengthening the Reporting of Observational Studies in Epidemiology (STROBE) statement [20] and the Consensus-based Checklist for Reporting of Survey Studies (CROSS) [21]. Preparatory discussions were initiated with stakeholders in targeted school districts prior to study commencement.

### 2.2. Data Collection Methods

The primary data collection tool was a novel single-step online survey (See Appendix A). The seven most frequently cited standardized tools (i.e., the Texas School Survey [22], Monitoring the Future [23], the National Youth Tobacco Survey [12], the Youth Risk Behavior Survey [24], the National Survey of Drug Use and Health [25], the National Health Interview Survey for Young Adults [26], and the Population Assessment of Tobacco and Health [27]) were reviewed and collated into one instrument to remove redundancies and illicit greater specificity regarding vaping specifically to further validate combustible cigarette inferences. These standardized instruments elicit information on vaping, assume conclusions based on research conducted on smoking and traditional tobacco usage, and are collected by many agencies, but they lack perspective on the adolescent psyche in relation to health outcomes. A case review demonstrated little evidence specifically addressing how adolescents perceive the health risks associated with vaping. Surveys have the disadvantage of removing the possibility of longitudinal follow-up but can be achieved in a relatively short period of time for a low cost. While several instruments are currently validated, most make assumptions about vaping based on data about smoking, and further investigations are needed to identify whether key differences exist between the two.

This survey was created in Google Forms and consisted of 5 demographic questions to assess eligibility; 26 questions regarding vape use, perceptions, and access; 10 vape knowledge questions; and 10 additional demographic and household questions. The survey was designed to allow questions to be skipped without barriers to future questions to collect as much data as possible if respondents exited the survey early.

The primary outcome was vape use. Students who vape were identified via an affirmative response to the question, “Do you vape?” Students who do not vape were identified similarly through a negative response. Regular vaping was defined as those that answered “yes” to the question “do you vape?” and did not select the answer choice “don’t regularly” to the next question “if yes, how old were you when you first started vaping regularly?” Regular smoking was defined similarly. Regular vaping was a secondary outcome given the small sample size.

Attitudes and beliefs were evaluated between preidentified groups of interest based on concepts from the Health Belief Model. The Health Belief Model is based on the premise that individuals place value on avoiding illness, injury, or negative health outcomes as compared against the expectations that prescribing to or avoiding certain actions or behaviors may be preventative versus the internal perceived acceptability of those behaviors. Perception is the individual reflection on a person’s impressions of social acceptance, risk, and pleasure. The core constructs can be broken down based on perceived susceptibility (believed likelihood), severity (believed seriousness), threat (believed susceptibility multiplied by believed severity), benefits (believed positive features), barriers (believed obstacles), cues to action (triggers), and self-efficacy (confidence) [28]. Perceived susceptibility was based on response to the question, “Do you think vaping is dangerous?”, and perceived severity was based on response to the question, “How likely are YOU to get sick from regular vaping?” (Appendix A, Vaping Survey, Q7 and Q8). The perceived threat was calculated by taking the means of the perceived susceptibility and perceived severity scores.

A secondary categorization was completed for analysis, distinguishing age into three groups: adolescents from 12 to 14 years, adolescents from 15 to 17 years, and adolescents from 18 to 20 years. The rationale behind the use of the three groups included enabling the evaluation in totality and categorically by criteria, highlighting early, middle, and late adolescence. This afforded simplicity yet still allowed for consideration of variable individual transition points throughout adolescence, which is often subject to sociocultural interpretation [29]. While participants in Group C might have fallen outside the primary age range of middle and high school adolescents, the similarities in age and developmental status were considered to mitigate the risk of missing eligible older students and allowed for a greater degree of generalization.

### 2.3. Sample Characteristics

The study population was comprised of adolescents aged 12 to 20 years in middle and high school. San Antonio is a major metropolitan area in South-Central Texas (Alamo Region [30]), with approximately two and a half million predominantly Hispanic (64%) residents, making it the third-largest city in Texas and the seventh-largest city in the United States [31]. Adolescents were identified from a total of eleven schools in six districts representing large, middle, and small school districts of rural and urban geographies. Economically challenged, low-income inner-city regions were also included in the sample planning process, allowing for the evaluation of socioeconomic health disparities. There were over 15,000 students enrolled in the targeted school districts at the time at which the survey was administered.

The chosen study population was mapped closely with but expanded upon the population categories of the Population Assessment of Tobacco and Health (PATH) project, which had thousands of respondents and focused on adolescents (12 to 17 years of age) [32]. A secondary exploratory group incorporating those outside this age range (18 to 20 years of age) but who were below the minimum regulatorily restricted purchase and consumption age of 21 years was also included. A three-year margin was implemented to identify Group A (12 to 14 years), Group B (15 to 17 years), and Group C (18 to 20 years) for incremental evaluation purposes, deviating slightly from the developmental phases of adolescence of Chulani and Gordon [33]. Baseline demographics were selected to be as comprehensive as possible, identify potential confounders and trends, and maximize participant anonymity.

### 2.4. Survey Administration

The distribution of the survey initially commenced with communications alerting stakeholders of the active collection status via e-mail. The stakeholders were asked to distribute the survey via their respective networks to solicit voluntary participation from adolescents. A brief overview of the study and a survey hyperlink were provided in the initial messages. A QR code was developed and added to the brief overview with the hyperlink within a few days of the initial administration and recirculated to encourage greater participation via ease of access.

Potential participants were contacted by social media, text, and e-mail. Furthermore, word-of-mouth distribution expanded the digital footprint. Participation was completely voluntary for both participants and administrators, leading to a small sample. Various communication channels used to reach adolescents regarding vaping have been preliminarily explored by others [34]. Text messages have been shown to be well received among adolescents when distributing brain and chemical information and nicotine exposure warnings. Social media (e.g., Twitter, Facebook, Instagram) was also used to distribute links to the survey and QR code. Friends and family were encouraged to circulate the survey within their networks without any direct knowledge, access, or solicitation by the principal investigator of any participant for anonymity and confidentiality purposes. Efforts were made to be inclusive in recruitment and participation. No participants were excluded from the study based on sex, race, ethnicity, or social standing.

### 2.5. Ethical Considerations

This study was reviewed and approved by the University of Texas Health San Antonio Institutional Review Board (protocol HSC20210154HU). Participant consent was obtained on the first page of the survey, which contained an instruction sheet including language for voluntary, anonymous participation, and withdrawal. Parental consent was waived as it was determined that participant anonymity would be better protected without parental consent.

### 2.6. Statistical Analysis

All statistical analyses were performed using JMP Pro 14 Statistical Software (SAS Institute, Cary, NC, USA). As the participants could exit at any time, all data collected prior to discontinuation were analyzed. Respondents failing to complete a question or response were removed from analysis for that question only.

Descriptive statistics were used to summarize participant demographic and household characteristics. Next, these characteristics and vaping severity, susceptibility, and perceived threat were compared between students who vape and students who do not vape, using bivariable statistical tests (the chi-squared test, Fisher’s exact test, *t*-test, or the Wilcoxon rank-sum test, as appropriate). Additional survey question responses were presented descriptively as the frequency and percentage of students.

## 3. Results

This study had 302 survey respondents from eleven schools across six school districts in South-Central Texas; 267 respondents met the study’s eligibility criteria and were included in this analysis. Respondents were excluded for the following reasons: having previously taken the survey (*n* = 8), age < 12 or >20 (*n* = 7), grade < 6th or >12th (*n* = 17), and non-attendance at the surveyed public middle or high schools (*n* = 3).

### 3.1. Baseline Characteristics and Vape Use

Table 1 contains the demographics of the eligible survey respondents. The respondents were from the following age groups: 12–14 years of age (14%), 15–17 years of age (68%), and 18–20 years of age (18%). Most were female (61%), from grades 9 to 12 (92%), earned “A’s” or “A’s & B’s” in school (77%), were Non-Hispanic White (45%) or Hispanic (41%), spoke English as their predominant language (87%), had both parents in the home (75%), had siblings in the home (76%), and earned an individual income (68%).

A total of 19 respondents (7%) reported that they vape. Six individuals (32%) stated they have vaped ≥4 times per week in the past year. Five individuals (26%) stated they have vaped an unknown amount, but >1 time per week in the past year. Two (11%) stated they have vaped > 1 time per month in the last year. Two (11%) stated they have vaped ≥1 but <3 times per month in the last year. Two (11%) stated they have vaped >5 but <12 times in the last year. Two (11%) stated they have vaped <5 times the last year. Students who vape were significantly more likely to speak English or other languages (*p* = 0.01 for Predominant Language) but less likely to speak Spanish or have or have siblings in the home (*p* = 0.04 for Siblings in the Home). Although not statistically significant, there was a higher percentage in the vape group compared to the non-vape group for males, 15–17-year-olds, 10th graders, and those with an individual income.

Figure 1 depicts the age at first vape and experimentation with smoking (Appendix A, Vaping Survey, Q1–2, 4–5). Sixteen individuals (6%, *n* = 264) stated that they vape regularly. In total, 69 (26%, *n* = 264) individuals stated they had tried vaping; note that 3 out of 267 individuals did not answer this question. Five individuals (2%, *n* = 262) stated they smoke regularly. Twenty-nine (11%, *n* = 256) individuals stated they have tried smoking. Most respondents (92%) tried vaping between 12 and 17 years of age. In contrast, most respondents (62%) tried smoking between <8 and 14 years of age. Similarly, most respondents (69%) started vaping regularly between 12 and 17 years of age, while 60% of respondents started smoking regularly before 8 years of age. Four regular smokers (80%, *n* = 5) stated they also vape regularly. Six individuals who vape (32%, *n* = 19) stated they do not smoke cigarettes regularly but have tried smoking cigarettes. Twelve individuals who have tried smoking cigarettes (41%, *n* = 29) stated they have also tried vaping.

### 3.2. Perceptions

Most (63%) respondents did not believe smoking and vaping were synonymous (Appendix A, Vaping Survey, Q3). Furthermore, 59% stated that vaping was not tobacco use (Appendix A, Vaping Survey, Q3a). While the majority (74%) felt that their parents would strongly disapprove of their vaping, they were less certain their friends would strongly disapprove (27%). The respondents also felt that at least some of their friends vaped (76%).

Figure 2 depicts the perceived threat of negative health outcomes due to vaping. The respondents who reported that they vape had significantly lower severity (72% vs. 97%, *p* < 0.01), susceptibility (39% vs. 82%, *p* < 0.01), and perceived threat (56% vs. 90%, *p* < 0.01) scores than those who reported that they do not vape.

### 3.3. Access

Figure 3 describes the ease of vape procurement and use (Appendix A, Vaping Survey, Q13–15). Most respondents (64%) said it was very easy (21%), easy (24%), or slightly easy (19%) to vape at school. Over half (61%) said it was very easy to vape before or after school. Finally, most respondents (70%) said it was very easy (21%), easy (29%), or slightly easy (20%) to obtain vape supplies.

Figure 4 depicts sources of vapes and supplies (Appendix A, Vaping Survey, Q16). The respondents could select more than one category. The most common sources included friends (74%), vape shops (57%), and gas stations (51%). Thirty-two individuals (14%, *n* = 235) stated they had been inside a vape shop, with 88% not having been asked for age verification via identification (ID) upon entry (Appendix A, Vaping Survey, Q19). Twelve (38%, *n* = 32) stated they tried to buy something, and 58% were not asked for ID at the point of sale.

## 4. Discussion

This study used a novel survey to collect information about the use of vaping, perceptions of vaping, and access to vaping among a sample of adolescents in South-Central Texas. Because of social media, companies with global reach, and international campaigns, we now live in a global society where adolescents in Texas can easily communicate with and be influenced by adolescents from around the world (and vice versa). Because of this, it is important to know about perceptions of vaping, access to vaping, and the use vaping for different populations around the world when making laws and policies relating to vaping. While the percentages of respondents who vape regularly or have tried vaping were relatively low, important trends were identified that may be leveraged for the development of additional standardized surveys and public health interventions. 

The majority of those surveyed did not think vaping and smoking cigarettes are the same thing. Most respondents stated they did not see vaping as tobacco use or were unsure. This is important as most of the current instruments available rely on smoking questions to predict and assume correlations with vaping. Additionally, vaping appears to be a significant contributing factor to increased rates of adolescent nicotine and tobacco use [35]. Manufacturers are subtlety marketing the distinctions between cigarette and vape use, touting vaping as safer than smoking. Previous research suggests that in the United States, 12th graders’ reasons for vaping included experimentation, entertainment, and cigarette replacement [36]. However, most users in this study reported vaping for reasons not related to cigarette use. Next, most respondents felt their parents would strongly disapprove of their vaping, but far fewer were certain their friends would strongly disapprove. Additionally, most respondents felt that at least some of their friends vaped. This suggests social acceptance, which might lead to acculturalization. This is significant based on social cognitive theory in which in summary, the more exposure or observations any member of a social group has to habitual behavior, the more likely they are to initiate that modeled behavior. Therefore, the perception that so many members of the social environment vape might indicate an increased risk for vaping in the expanded social network.

Interestingly, the age at which the respondents initiated smoking was younger than the age of initiation for those that had tried vaping. The modeling of parental behavior when smoking is observed in the home might account for this observation. For the same reason, accessibility, when cigarettes might be taken from adults without their knowledge, might be another reason. The dual use of e-cigarettes and traditional cigarettes was also seen in this sample, with most respondents who identified as regular smokers also identifying as regular vapers. Additionally, a large portion of those who have experimented with traditional cigarettes have also experimented with e-cigarettes. Smoking might be a gateway to vaping as much as vaping has been observed to be a gateway to smoking. More research with a larger sample size is needed as these findings go against the common argument that vaping might lead to smoking.

Respondents found it easy, to some degree, to vape before, during, and after school. Vapes are easy to conceal, making them difficult to detect. This complicates the enforcement of school anti-vaping policies by teachers and administrators. The findings noted in the results are significant because they signal relatively few deterrents for adolescents to abstain or avoid detection while vaping throughout their routine activities, as encountered during a normal day. Additionally, adolescents often acquire vapes and associated supplies from those they know, like friends, siblings, or even their parents. However, around two-thirds of adolescents stated they would not spend any money on vaping. This might infer that the cost is considered too high. Regulations imposing taxes on vaping products increase the overall purchase price. This might be a sign that at least some of the regulatory strategies are working to deter the use of vapes by adolescents. Consistent with the literature [37], the survey demonstrated that adolescents obtain vape products from numerous sources. It was noted in the literature that there is a higher density of vape shops and retail locations selling vape supplies in urban areas targeting minority and other socioeconomically disadvantaged groups.

This study has potential limitations. First, there was significant non-response; therefore, the respondent population may not fully represent the target population. Given the relatively small sample size, few regular vapers and smokers were identified, which may have resulted in underpowered comparisons in characteristics and outcomes between these groups. The sample size also limited our ability to identify precise age group tends, especially among the younger (12–14 years) and older (18–20 years) subgroups. Additional limitations include a lack of validation due to the novelty of the survey, a lack of Spanish translation, potential inaccessibility for those without internet services, an inability to follow-up or use Delphi-type longitudinal research (group opinion), a lack of comparison of dual users of e-cigarettes and traditional cigarettes in figures, the potential to miss a priority population, or possible inaccurate responses. Challenges were encountered due to COVID-19 protocols, year-end standardized testing, and changes in administrative personnel within the targeted school districts, probably reducing the full potential number of participants over the enrollment period.

Despite these limitations, this study was strengthened using primary data collected using a novel single-step, vaping-focused online survey eliciting the attitudes and beliefs of adolescents. Other strengths of the survey included its ease of administration, anonymity, reduced interrater variability and social desirability bias, and low cost. Future versions would benefit from a more structured recruitment process to capitalize on the full support of school administrators for the project. Efforts should be made in future work to recruit a larger sample comprising those who vape or smoke.

## 5. Conclusions

In this cross-sectional survey study of adolescents in South-Central Texas, the prevalence of regular vaping was relatively low, though most respondents reported greater ease of access to vaping and a lower perceived health threat from vaping compared to traditional tobacco smoking. The prevalence of vaping among adolescents continues to rise, complicated by the fact the age of initiation continues to decline. Risky health behaviors associated with vaping should not be assumed to be causative in nature. However, this study suggests risky health behaviors might be predictive in nature when evaluated as cumulative markers associated with the potential to vape. By using a proactive analysis model of adolescents’ perceptions, defined as attitudes and beliefs, researchers can evaluate the potential, or risk, for vaping. We must gain a better understanding of adolescents’ perceptions of vaping, access to vaping, and use of vaping if we hope to identify potential ways to combat the health burdens associated with vaping. Merely informing adolescents about safety concerns is doing little to discourage use. Therefore, more must be done to understand their vulnerability and risk tolerance for unhealthy behavior. Goals should include improving decision making and demonstrative habitual behaviors in the hopes of reducing the incidence and prevalence of negative health outcomes associated with the use of e-cigarettes or vaping products.

## Figures and Tables

**Figure 1 ijerph-20-06766-f001:**
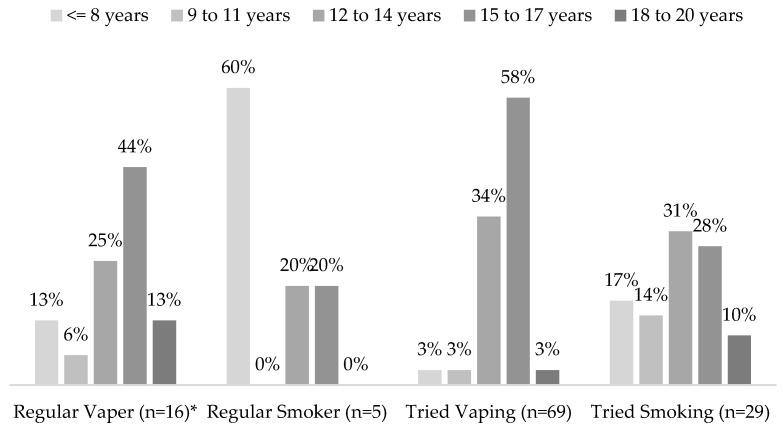
Age of first vape and experimentation with smoking. * *n* = 19 (three respondents answered “Don’t regularly” in response to the question “If yes, how old were you when you first started vaping regularly?” (Appendix A, Vaping Survey, Q1b)).

**Figure 2 ijerph-20-06766-f002:**
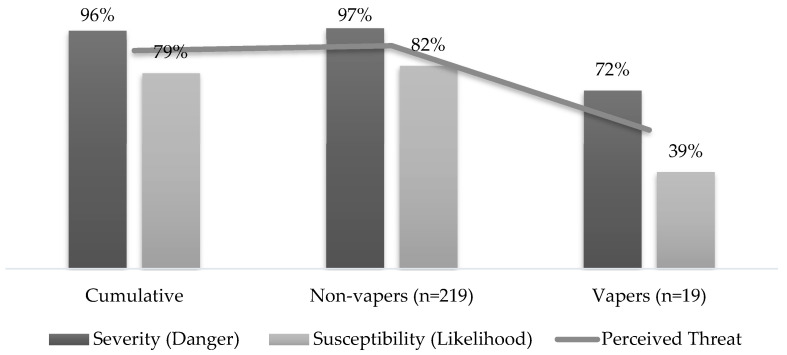
Perceived threat of negative health outcomes due to vaping, *n* = 238.

**Figure 3 ijerph-20-06766-f003:**
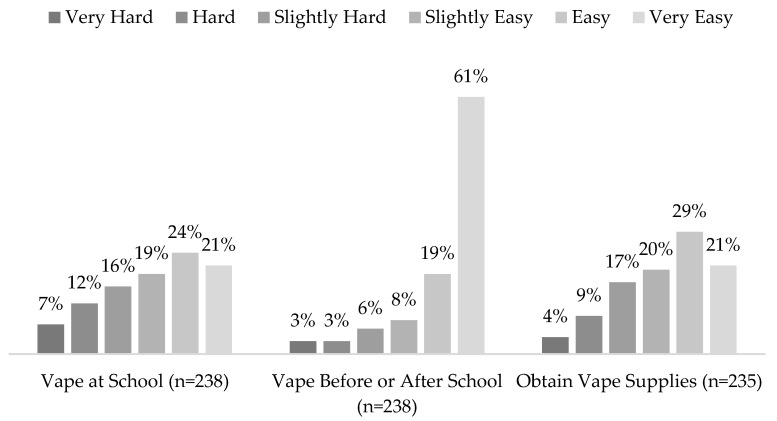
Ease of vape procurement and use.

**Figure 4 ijerph-20-06766-f004:**
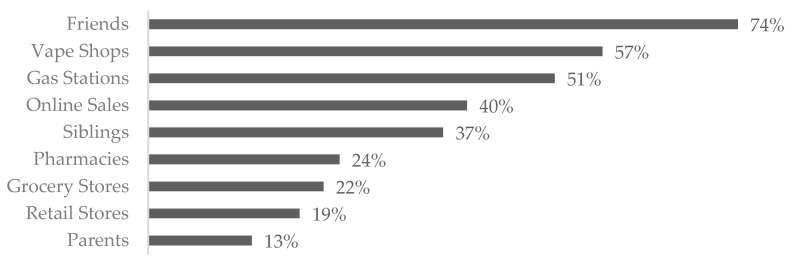
Sources of vapes and supplies, *n* = 222.

**Table 1 ijerph-20-06766-t001:** Survey respondent characteristics.

Characteristic	Overall (*n* = 267)		Students Who Do Not Vape (*n* = 248)		Students Who Vape (*n* = 19) *		*p*-Value
	%	Count	%	Count	%	Count	
Sex	*n* = 253		*n* = 234		*n* = 19		0.49
Female	61%	153	61%	143	53%	10	
Male	40%	100	39%	91	47%	9	
Age	*n* = 264		*n* = 245		*n* = 19		0.19
12–14 (Group A)	14%	37	15%	37	0%	0	
15–17 (Group B)	68%	179	67%	164	79%	15	
18–20 (Group C)	18%	48	18%	44	21%	4	
Grade	*n* = 264		*n* = 245		*n* = 19		0.24
12th	25%	66	25%	61	26%	5	
11th	17%	46	18%	44	11%	2	
10th	24%	62	22%	53	47%	9	
9th	27%	70	27%	67	16%	3	
8th	<1%	2	<1%	2	0%	0	
7th	2%	4	2%	4	0%	0	
6th	5%	14	6%	14	0%	0	
Predominant Grades	*n* = 187		*n* = 172		*n* = 15		0.64
A’s	39%	72	40%	68	27%	4	
A’s & B’s	39%	72	38%	65	47%	7	
Other	23%	43	23%	39	27%	4	
Race	*n* = 184		*n* = 169		*n* = 15		0.42
White (non-Hispanic)	45%	82	44%	75	47%	7	
White (Hispanic)	26%	48	26%	44	27%	4	
Hispanic	15%	27	16%	27	0%	0	
Other	15%	27	14%	23	27%	4	
Predominant Language	*n* = 186		*n* = 171		*n* = 15		0.01
English	87%	162	87%	148	93%	14	
Spanish	9%	17	10%	17	0%	0	
Other	4%	7	4%	6	7%	1	
Both Parents in Home	*n* = 187		*n* = 172		*n* = 15		0.69
Yes	75%	141	77%	132	60%	9	
No	25%	46	23%	40	40%	6	
Siblings in Home	*n* = 185		*n* = 170		*n* = 15		0.04
Yes	76%	140	78%	132	53%	8	
No	24%	45	22%	38	47%	7	
Individual Income	*n* = 187		*n* = 172		*n* = 15		0.24
Yes	68%	127	67%	115	80%	12	
No	32%	60	33%	57	20%	3	

* Defined as those who answered yes to the question “Do you vape?”.

## Data Availability

De-identified study data will be made available upon request to the corresponding author, B.A.G.

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
