# Peer review of "A Survey of Vaping Use, Perceptions, and Access in Adolescents from South-Central Texas Schools"

_ijerph, 2023, doi:10.3390/ijerph20186766_

Round 1
Reviewer 1 Report (Previous Reviewer 1)
All the previous comments have been addressed however, the ethical concern of waiving parental consent may be a contentious issue. Nevertheless, considering the importance of under-age access to vape it is an important public health. this is public health interest is an acceptable exclusion of ethics of consent procedure. The authors present mainly descriptive data. There is somewhat lack inferences at are factors associated with easy of access to those use VAPE with knowledge perceptions etc which needs to be explored further.
Author Response
Thank you for your helpful comments and suggestions. We have attached a detailed, item-by-item report to respond to each of these. The manuscript is better for having incorporated your suggestions. Thank you.

Reviewer 2 Report (Previous Reviewer 2)
The study applied a novel electronic survey to assess use, perceptions, and access of vaping among 267 adolescents in eleven schools in South-Central Texas in 2021. The descriptive and bivariate statistical results were reported, however, there are many errors. My 8 comments were attached.
1. In line 23, “A total of 267 respondents were included; 20% had tried vaping.” However, the number of tried vapers is 69 based on the results in Figure 1, and the percentage should be 26% (=69/267) rather than 20%. I cannot find any data in the manuscript to support 20%. Please revise.
2. Missing label for the first two columns of the results in Table 1. Please add a label between “Characteristic” and “Students who do not vape”.
3. In lines 202-203, the proportion of Non-Hispanic White should be 45% rather than 44%, and the proportion of having both parents in the home should be 75% rather than 77% based on the results shown in Table 1. Please check.
4. In lines 205-206, “Students who vape were significantly more likely to speak English (p=0.01) and have siblings in the home (p=0.04)” should be revised as “Students who vape were significantly more likely to speak English or other languages (p=0.01 for Predominant Language) but less likely to speak Spanish or have siblings in the home (p=0.04 for Siblings in Home)”.
5. In lines 208-209, “students with two parents in the home,” should be removed from the sentence, since the vape group has lower percentage (60%) compared with the non-vape group (77%) for this variable based on the results in Table 1.
6. In lines 213-214, “Sixty-nine (11%, N=264) individuals stated they have tried vaping.” The percentage should be 26% (=69/264) rather than 11%.
7. Due to the small number of total vapers (n=19), please explain clearly the number of vapers and non-vapers in Figure 2 when the total sample size reduced from 267 (Table 1) to 238. If the number of vapers is less than 10 in Figure 2, I doubt the meaning of comparisons.
8. Are there any dual users of cigarettes and e-cigarettes in your samples? When doing the comparisons in Figure 1, dual users may affect the patterns for vaping and smoking. Please explain more about this.
Author Response
Thank you for your helpful comments and suggestions. We have attached a detailed, item-by-item report to respond to each of these. The manuscript is better for having incorporated your suggestions. Thank you.

Reviewer 3 Report (Previous Reviewer 3)
I have thoroughly reviewed the revised manuscript and the authors’ responses to all 3 reviewers. Overall, the authors were responsive to a challenging set of reviews for this manuscript reporting on a cross-sectional design with data collected on vaping perceptions, access, and use among individuals from San Antonio, Texas between the ages of 12 and 20 years. The manuscript content is far clearer and markedly improved throughout. However, enthusiasm continues to be reduced by several data-related limitations, including the large amount of missing data among many respondents and the extremely high level of survey non-response. It is also still unclear exactly how many students were “approached” (or emailed/texted/contacted through social media to complete the survey link) and did not attempt to open the survey. Usually research with web-based surveys has the capability to report the number of respondents who clicked on a link but did not enter any data. Delineating non-responders to the #/% who “opened the survey link but did not provide any data” and the #/% who “did not open the survey link” would be enlightening. Given that 15,000 students were enrolled in this school system, and fewer than 300 students completed some portion of the survey, it seems extremely unlikely that these data are even moderately representative of the intended population. Calling this research a “pilot” and reporting this as a limitation does not fully mitigate these drawbacks.
Additional feedback:
1) Do the authors have access to any data on non-responders for comparison to data from responders? Short of these types of comparison, it’s very difficult to know how precise/imprecise the reported estimates may be.
2) Based on the points outlined above, this reviewer disagrees that “feasibility of disseminating” (as stated in the Discussion) was demonstrated. More work needs to be piloted to increase the response rate.
Good luck!
Author Response
Thank you for your helpful comments and suggestions. We have attached a detailed, item-by-item report to respond to each of these. The manuscript is better for having incorporated your suggestions. Thank you.

This manuscript is a resubmission of an earlier submission. The following is a list of the peer review reports and author responses from that submission.
Round 1
Reviewer 1 Report
The manuscript sounded like an interesting topic from the title but when reading the full manuscript there are many writing issues that need to be fixed before can be further considered.
The methods has to reorgniased as per the STROBE and other guidelines for survey research. there is a very long discussion about participant selections and justifications which should be indeed participant criteria for inclusion or exclusion. Many statements about surveys methods ate explained here. under design and setting they describe survey questionnaire developments. It should be a separate section study measures that do not define all measures shown in the results such as daily or non-daily vapers, smokers, etc.
Why was the sample size not calculated:? looks like used a convenient sample why not specify this? Was there any non-response? Why was parental consent not taken if the participants were younger than the legal age to provide consent? Even for others why was the consent procedure overlooked? needs a much stronger justification.
In the results, there are several figures that have comparisons, but the listed statistical tests and their significance is not shown. please be specific about which statistical tests were applied where and how these were suitable for these comparisons.
Discussion is very generic, without a focus on what was found? conclusions were not succinct based on results and these run into several paragraphs.
suitable
Reviewer 2 Report
The manuscript applied a novel electronic survey to assess perceptions, access, and use of vaping among 302 adolescents (ages 12-20 years) in eleven schools in South Texas from May to August 2021. It’s a great study, however, the definitions and results are not clear for me. My 8 comments were attached.

Reviewer 3 Report
See PDF.
